# Investigation of PACAP38 and PAC1 Receptor Expression in Human Retinoblastoma and the Effect of PACAP38 Administration on Human Y-79 Retinoblastoma Cells

**DOI:** 10.3390/life14020185

**Published:** 2024-01-26

**Authors:** Dénes Tóth, Eszter Fábián, Edina Szabó, Evelin Patkó, Viktória Vicena, Alexandra Váczy, Tamás Atlasz, Tamás Tornóczky, Dóra Reglődi

**Affiliations:** 1Department of Forensic Medicine, University of Pécs Medical School, Szigeti út 12, 7624 Pecs, Hungary; 2Department of Anatomy, HUN-REN-PTE PACAP Research Team, Centre for Neuroscience, University of Pécs Medical School, Szigeti út 12, 7624 Pecs, Hungary; eszter.fabian@aok.pte.hu (E.F.); szaboedina90@gmail.com (E.S.); evelin.patko@gmail.com (E.P.); vicena.viktoria@gmail.com (V.V.); vaczyalexandra@gmail.com (A.V.); attam@gamma.ttk.pte.hu (T.A.); dora.reglodi@aok.pte.hu (D.R.); 3Department of Sportbiology, University of Pécs, Ifjúság út 6, 7624 Pecs, Hungary; 4Department of Pathology, University of Pécs Medical School and Clinical Center, 7624 Pecs, Hungary; tornoczki.tamas@pte.hu

**Keywords:** PACAP, retinoblastoma, enucleation, immunohistochemistry, cell survival, cell viability

## Abstract

Retinoblastoma represents the most prevalent malignant neoplasm affecting the eyes in childhood. The clear-cut origin of retinoblastoma has not yet been determined; however, based on experiments, it has been suggested that *RB1* loss in cone photoreceptors causes retinoblastoma. Pituitary adenylate-cyclase activating polypeptide (PACAP) is a pleiotropic neuropeptide which has been shown to be affected in certain tumorous transformations, such as breast, lung, kidney, pancreatic, colon, and endocrine cancers. This study aimed to investigate potential changes in both PACAP38 and PAC1 receptor (PAC1R) expression in human retinoblastoma and the effect of PACAP38 administration on the survival of a human retinoblastoma cell line (Y-79). We analyzed human enucleation specimens removed because of retinoblastoma for PACAP38 and PAC1R immunostaining and the effect of PACAP38 on the survival of the Y-79 cell line. We described for the first time that human retinoblastoma cells from patients showed only perinuclear, dot-like immunopositivity for both PACAP38 and PAC1R, irrespective of laterality, genetic background, or histopathological features. Nanomolar (100 nM and 500 nM) PACAP38 concentrations had no effect on the viability of Y-79 cells, while micromolar (2 µM and 6 µM) PACAP38 significantly decreased tumor cell viability. These findings, along with general observations from animal studies showing that PACAP38 has strong anti-apoptotic, anti-inflammatory, and antioxidant effects on ocular tissues, together suggest that PACAP38 and its analogs are promising candidates in retinoblastoma therapy.

## 1. Introduction

Retinoblastoma, acknowledged as one of the leading childhood malignancies, is the most prevalent primary intraocular neoplasm that tends to manifest in early childhood [1]. The worldwide occurrence of retinoblastoma is approximately 1 in 16,000–18,000 live births [2]. However, there are variations in incidence among countries, regions, and ethnic groups, with almost similar incidence rates for males and females [3]. Developing countries with high birth rates have the highest mortality (40–70%) compared with Europe, the USA, and Canada (3–5%) [4]. The causes of these differences in mortality include delays in diagnosis, advanced stages of the disease, lack of access to health care systems, and other socioeconomic factors [5].

Biallelic loss-of-function mutations in the tumor suppressor gene RB1, located on chromosomal region 13q14, account for the initiation of retinoblastoma in 95% of cases [6]. Individuals with germline mutation (first hit) require only one additional hit (acquired somatic mutation). Therefore, heritable retinoblastoma occurs at a younger age and is usually multifocal or bilateral. In contrast, most sporadic cases are unilateral as a result of mutation in both alleles of *RB1* [7]. In rare cases, sporadic retinoblastoma could develop in the absence of *RB1* mutation as a consequence of the somatic amplification of the *MYCN* gene [8]. The exact cellular origin of retinoblastoma is not clear yet [9,10]. Based on live imaging data of early tumors from patients’ eyes, the inner nuclear layer of the retina seemed to be the origin [11], but based on experiments, it has been suggested that human retinoblastoma arise from differentiating cones as it was found that retinoblastoma cells exhibit multiple elements of the cone precursor signaling circuitry and depend on this circuitry for their proliferation, survival [12], and *RB1* knockdown-induced human cone precursor proliferation [13]. Distinctions in molecular, clinical, and histopathological aspects among RB1−/− tumors reveal a progression marked by a loss of differentiation and a decline in the photoreceptor expression signature [14]. Human cone precursor maturation’s unique cell-signaling circuitry makes them sensitive to *RB1* loss, leading to proliferation and lesion formation resembling retinoma and retinoblastoma [15]. A human Rb organoid model also identified maturing cone precursors as the origin of human retinoblastoma [16]. A multi-omics approach identified two retinoblastoma molecular subtypes expressing cone markers [17]. Interestingly, *RB1* loss was observed to induce proliferation and tumorigenesis in maturing cone precursors, while it was found that the somatic amplification of the *MYCN* gene induced proliferation in immature cone precursors [18].

Retinoblastoma is characterized primarily by leukocoria, with subsequent symptoms including strabismus, a red and painful eye, impaired visual tracking, and vision loss. Diagnosis is usually clinical (ophthalmoscopy) combined with additional examinations like ultrasonography, computer tomography, or magnetic resonance imaging [5]. Retinoblastoma is usually white and has a brain-like appearance with pale areas of calcification and yellowish necrotic zones. The histopathological characteristics of retinoblastoma encompass small hyperchromatic cells exhibiting a high nuclear-to-cytoplasmic ratio, accompanied by areas of variable-sized necrosis and dystrophic calcification [19]. The level of retinal differentiation in retinoblastoma varies and correlates negatively with the age of a child [20]. Differentiated tumors include fleurettes, representing advanced photoreceptor differentiation; Flexner–Wintersteiner rosettes, showing early retinal differentiation; and Homer Wright rosettes, exhibiting primitive neuroblastic differentiation [21,22].

The current therapy for retinoblastoma depends on the time of disease detection and socioeconomic factors. In low- and middle-income countries, therapy is aimed at protecting patients’ lives due to enucleation, followed by salvage of the globe and vision. Unfortunately, enucleation is still the most frequent choice worldwide [23].

An evolutionarily conserved neuropeptide, pituitary adenylate-cyclase activating polypeptide (PACAP), is recognized for its multifunctionality and pleiotropy. Its established anti-apoptotic, anti-inflammatory, and antioxidant effects [24,25] are attributed to two functionally active isoforms: PACAP38 [26] and PACAP27 [27]. The latter represents 10% of the total PACAP in the body, while PACAP38 is the predominant form in mammalian tissues. PACAP acts on two nonspecific and one specific G-protein-coupled receptors. Highlighting its selective nature, the PAC1 receptor (PAC1R) responds exclusively to PACAP, in contrast to the shared VPAC1 and VPAC2 receptors that accommodate both PACAP and VIP. Across the central and peripheral nervous systems, as well as in peripheral organs, PACAP and PAC1R display broad and widespread expression [28].

The current understanding of the presence, distribution, and functional aspects of PACAP and its specific receptor in the human eye is limited. In a previous study, Olianas and coauthors [29] provided evidence showcasing that PACAP has the capacity to elevate cAMP levels in the retina. Additionally, from retinal homogenates obtained from human fetuses, they successfully identified the mRNA for both PACAP and its receptors. The first detailed exploration of the distribution of PACAP38 and PAC1R within the normal human eye was conducted in 2022 [30]. Corneal epithelial and endothelial cells, the iris (both muscle and stroma), the ciliary body, several retinal layers, and the optic nerve showed immunopositivity for both PACAP38 and PAC1R. Regarding the retina, the pigment epithelial layer—forming from the original outer eyecup layer—showed strong expression for PACAP38 and PAC1R. Weak or no immunostaining was observed in the outer nuclear and plexiform layers where rods and cones are located. In the majority of cases, the inner nuclear layer exhibited a markedly positive expression for both the peptide and PAC1R. The inner plexiform layer had strong PACAP38 and PAC1R immunoreactivity. Interestingly, a specific staining pattern was observed in the ganglion cell layer, where some ganglion cells showed very strong, and others showed negative, immunosignals [30]. PACAP takes part in a wide range of physiological [31,32,33] and pathological processes [34,35]. In vitro studies with human corneal endothelial cells and pigment epithelial cells proved that PACAP stimulates adenylate cyclase and various intracellular signaling pathways to protect the cells against various noxas, including hyperglycemia, oxidative stress, and growth factor deprivation [36,37,38,39,40,41]. In vivo studies showed the protective effect of PACAP in different types of retinopathies [40,42,43,44], glaucoma [45], and injuries [46,47]. PACAP also affects cellular differentiation [48], cell division, cell cycle, and cell death [49]. As PACAP regulates almost every aspect of stem cell physiology [50], it is not surprising that PACAP and its receptors were detected in numerous human cancer types [51,52,53]. Many different tumors show over- or under-expression of the PACAPergic system. The activation of PACAP receptors in specific neoplasms can lead to growth stimulation, whereas in others, it results in inhibitory effects. These effects depend on numerous factors, like the origin and type of the tumor, the stage of differentiation, and the tumoral environment [52,54]. As PACAP is shown to be involved in certain tumorous transformations [55] and changes in PACAP and PAC1R can be detected under pathological conditions [34,35], this study aimed to investigate potential alterations in the expression of PACAP38 and PAC1R within tumor tissue samples obtained from enucleation specimens of retinoblastoma patients and the effect of PACAP38 treatment on the survival of human Y-79 retinoblastoma cells.

## 2. Materials and Methods

### 2.1. Human Eyes

This study was conducted according to the ethical standards specified in the Declaration of Helsinki, along with due consideration of the corresponding regulations within Hungarian law. The collection of all samples strictly followed a protocol that received approval from the Institutional Ethics Committee at the University of Pecs (9188-PTE 2022; approval date: 10 June 2022). The identification of cases involved a comprehensive search through the pathological records of the Department of Pathology at the University of Pécs Medical School, covering the period from January 2001 to December 2017. This search specifically targeted enucleation specimens carrying a histopathological diagnosis of retinoblastoma. Patient medical records were reviewed for demographic information and clinical findings. The staging was conducted following the eighth edition of the Cancer Staging Manual of the American Joint Committee on Cancer (AJCC) [22]. In brief, pT1 denotes intraocular tumor(s) without local invasion, focal choroidal invasion, or involvement of the optic nerve head either pre- or intralaminarly. pT2 corresponds to intraocular tumor(s) with local invasion, pT3 indicates intraocular tumor(s) with substantial local invasion, and pT4 signifies the presence of extraocular tumor(s) [22]. The level of differentiation (grading) was classified as follows: G1 corresponds to a tumor displaying areas of retinocytoma, characterized by fleurettes or neuronal differentiation, accounting for more than half of the tumor. G2 signifies a tumor featuring numerous rosettes, including Flexner–Wintersteiner or Homer Wright rosettes, constituting more than half of the tumor. G3 denotes a tumor with occasional rosettes (Flexner–Wintersteiner or Homer Wright rosettes), accounting for less than half of the tumor. G4 represents a tumor with poorly differentiated cells lacking rosettes or displaying extensive areas of anaplasia [56].

#### Immunohistochemistry

Tissue samples underwent fixation in a 10% neutral-buffered formalin solution, followed by dehydration through a series of graded ethanol solutions and eventual embedding in paraffin. Subsequently, sections with a three-micrometer thickness were cut using a rotational microtome (Microm HM 325, Thermo Scientific, Ltd., Waltham, MA, USA) and affixed onto coated glass slides. Following deparaffinization and rehydration through graded ethanol, the samples underwent pretreatment using a heat-induced epitope retrieval method. This involved subjecting the samples to a microwave oven for 15 min at 750 W in a 1 mM (pH = 6.0) citrate buffer. Subsequent to cooling at room temperature, the samples were washed in a pH 7.6 TRIS-buffered saline solution (TBS). Samples were then incubated for 1 h at room temperature with anti-PACAP38 (Cat. Nr. T-4473, BMA Biomedicals, Ltd., Augst, Switzerland, 1:500) and anti-PAC1-R antibody (Cat. Nr. AVR-003, Alomone Labs, Ltd., Jerusalem, Israel, 1:125). After washing in TBS, the sections were exposed to an HISTOLS-AP-R anti-rabbit alkaline phosphatase labeled detection system (Cat. Nr. 30,011.R500A, Histopathology, Ltd., Pécs, Hungary) for a 30 min incubation at room temperature. Subsequently, they underwent another round of TBS washing, and the enzymatic reaction was initiated in a dark environment using an HISTOLS Resistant AP-Red Chromogen/substrate System (Cat. Nr. 30,019K, Histopathology, Ltd., Pécs, Hungary). Following a 10 min incubation with the chromogen/substrate working solution, staining intensity was controlled under a light microscope. This chromogen substance was chosen because its magenta staining was visible in the pigmented cells. Counterstaining was performed with hematoxylin, and tap water was used for bluing. After drying, samples were dehydrated in graded ethanol and cleared in xylene before being mounted with a permanent mounting medium. For negative control, the primary antibody was replaced with TBS, which resulted in no staining. The healthy parts of the eye served as an internal positive control. Using a Panoramic MIDI II automatic digital slide scanner (3DHISTECH Ltd., Budapest, Hungary), the slides were thoroughly scanned, and images were captured through CaseViewer 2.3 software (3DHISTECH Ltd., Budapest, Hungary).

### 2.2. Cell Culture

Y-79 human retinoblastoma cells, sourced from the American Type Culture Collection (ATCC, Manassas, VA, USA), were cultivated in RPMI-1640 medium supplemented with 10% fetal bovine serum, 100 U/mL penicillin, and 100 μg/mL streptomycin. The culture was maintained in a humidified incubator at 37 °C with 5% CO_2_, and the culture medium was refreshed every second day. All cell culture reagents were procured from Sigma-Aldrich (St. Louis, MO, USA).

#### 2.2.1. Cell Viability Assay

A colorimetric assay for evaluating cell viability, an MTT (3-(4,5-dimethylthiazol-2-yl)-2,5-diphenyltetrazolium bromide) assay from Sigma-Aldrich (St. Louis, MO, USA), was utilized to investigate the effect of PACAP38. Cells were seeded in 96-well plates (4 × 10^3^ cells/well). Twelve wells served as control, and 100 µL of serum-free medium was added to these wells. Six wells in each column were treated with 10 µL 0.1 µM, 0.5 µM, 1 µM, 2 µM, and 6 µM PACAP38 (produced within the Department of Medical Chemistry at the University of Szeged, Szeged, Hungary) to observe the dose dependency. The wells were then filled with serum-free medium to a final volume of 100 µL. Following a 24 h incubation period, 10 µL of a 5 mg/mL MTT solution was added to each well, achieving a final concentration of 0.45 mg/mL. The plate underwent an additional 4 h incubation in a thermostat, after which the reduced formazan dye was dissolved using 100 µL DMSO (dimethyl-sulfoxide). After 30 min on a shaker, the absorbance was gauged at 630 nm using an ELISA reader (Dialab Ltd., Budapest, Hungary). The assay was performed in duplicate and repeated three times.

#### 2.2.2. Statistical Analysis

Statistical analysis was conducted using one-way ANOVA followed by Dunnett’s test with GraphPad Prism version 9.5.0 for Microsoft Windows (GraphPad Software LLC, San Diego, CA, USA). The presented data include means ± standard deviation (SD).

## 3. Results

### 3.1. Human Eyes

#### 3.1.1. Clinical Data

Seven children (one girl and six boys) were included in our study who underwent primary enucleation because of retinoblastoma. The mean age at enucleation was 16.3 ± 10.5 months (median: 16.6 months; range: 32.2). Except for two cases, the right eye was affected, and there were no bilateral cases. In three cases, the tumor site involved the superotemporal quadrant after sectioning, while in one case, the lesion was near the optic disc. In the remaining cases, the tumor occupied the entire eye, and the precise point of origin could not be determined. In all cases, with one exception, a monofocal tumor was observed, and in the case of multifocal retinoblastoma, genetic involvement could not be confirmed. However, in the monofocal case, familial clustering and an RB1 mutation were identified. Histomorphologically, in three cases, poorly differentiated cells without rosettes were found (G4), while in one case, a tumor with occasional Homer Wright rosettes (G3) was observed. In the remaining cases, a tumor with many rosettes was detected (G2). Among the latter, Homer Wright rosettes were visible in two cases, and Flexner–Wintersteiner rosettes were observed in one case. In all cases, various degrees of necrosis were present within the tumors, and except for two cases, calcifications were also observed. The Mib-1 (Ki-67) labeling index was relatively high, exceeding 50% in all cases (excluding two instances where such an examination was not conducted for technical reasons). Table 1 provides an overview of the primary demographic and clinical features, while Table 2 summarizes the main pathological findings of the cases.

#### 3.1.2. Immunohistochemistry

In accordance with our earlier observations [30], we found PACAP38 and PAC1R immunopositivity in disease-free areas of the eye, including the cornea (both epithelial and endothelial cells), the iris (stroma and muscles), the ciliary body, and different retinal layers (pigment epithelial layer, inner plexiform layer, and ganglion cell layer). These findings served as a positive internal control. Figure 1 includes representative pictures from the disease-free section of the eyes, which served as an internal control in the study.

Retinoblastoma cells showed only perinuclear, dot-like immunopositivity (black arrows) for both PACAP38 and PAC1R, irrespective of laterality and genetic background. In this immunopattern, there was no difference between the poorly differentiated samples or the different types of rosettes, regardless of the proportion of rosettes in the tumor (Figure 2).

### 3.2. Cell Culture

One-way analysis of variance indicated significant differences in cell survival among the treatment groups (F = 5.165, *p* = 0.0047). Dunnett’s multiple comparisons test revealed statistically significant differences between the control group and both the 2 µM PACAP38- (mean diff: 16.5, 95% CI [1.471, 31.52], *p* = 0.035) and the 6 µM PACAP38 (mean diff: 20.38, 95% CI [8.488, 32.26], *p* = 0.0053)-treated groups. Although the mean difference for the 0.5 µM PACAP38-treated group was similar to that of the 2 µM PACAP38-treated group (17.09 vs. 16.5), there were no statistically significant differences observed when compared to the control group (95% CI [−1.290, 35.48], *p* = 0.0653) (Figure 3).

## 4. Discussion

In the first part of our study, we analyzed human enucleation specimens removed because of retinoblastoma for PACAP38 and PAC1R immunostaining and described, for the first time, the distribution of PACAP38 and PAC1R expression in human retinoblastoma. We found PACAP38 and PAC1R immunopositivity in the tumor-free area of the eyes consistent with the results of the first description of the distribution of PACAP38 and PAC1R in the human eye [30] which, therefore, served as a positive internal control. In retinoblastoma, we observed only focal, perinuclear dot-like immunopositivity for both PACAP38 and PAC1R. There were no differences in the immunopatterns between the different histological features, i.e., the presence of different types of rosettes.

A broad spectrum of human cancers has been observed to express PACAP38 and PAC1R. Furthermore, certain tumors show alteration of the PACAPergic system compared to the normal tissue. In papillary thyroid carcinoma, the overexpression of PACAP38-positive cells was detected compared to normal thyroid glands, while colloid showed weaker or no staining pattern. Regarding PAC1R, tumor cells showed only minimal or no expression compared to the normal glands, where strong granular expression was present [57]. Similar overexpression was observed in the case of invasive ductal adenocarcinoma of the breast both for PACAP38 and PAC1R [58,59]. Lower PACAP38 and PAC1R immunosignals were detected in the case of pancreatic ductal adenocarcinoma and insulinoma compared to those in healthy pancreatic tissues [60,61]. In the case of non-small cell lung cancer, colon adenocarcinoma, and kidney tumor samples, a significantly lower level of PACAP38-like immunoreactivity was detected by RIA compared with that in normal healthy tissues [62,63]. Prostatic adenocarcinoma showed essentially preserved PACAP38 immunopatterns compared to those in normal prostatic glands [64], and no significant alterations in PACAP38-like immunoreactivity detected by RIA were found in cases of urinary bladder tumor samples, prostatic adenocarcinoma samples, or various types of testicular malignancies, like seminoma, embryonal carcinoma, yolk sac tumor, and teratoma [63]. Regarding the tumors mentioned above in the immunohistochemical studies, both the peptide and its specific receptor showed widespread expression, including membrane and intracytoplasmic PAC1R expression, and in none of these cases were they able to detect a pattern where PACAP38 and PAC1R were expressed only in a perinuclear dot-like pattern, as observed in our study.

Retinoblastoma differs from these types of tumors as the exact cellular origin of retinoblastoma is still controversial. For this reason, in our experiment, we were not able to compare the PACAP38 and PAC1R immunoprofiles of ‘normal’ and tumor tissues. We attempted to compare the immunopatterns observed in disease-free parts of the eyes within the inner nuclear layer, outer nuclear layer (housing the nuclei of photoreceptor cells), and the layer of rods and cones with the immunopattern observed in retinoblastoma. No or only faint PACAP38 and PAC1R immunosignals were identified in the inner nuclear layer, depending on the cell types located in this layer. Moreover, the same type of cells showed individually variable patterns. In this retinal layer, the faint immunopatterns of PACAP38 and PAC1R were morphologically (perinuclear dot-like positivity) similar to that observed in retinoblastoma. The same observations were made in the case of the outer nuclear layer, as well as in the layer of rods and cones. Since the originating cell type of retinoblastoma is unknown, and the cells potentially involved in this context also exhibited individual expression variations, we could not establish a parallel with the pattern observed in retinoblastoma.

Regarding human retinoblastoma cell lines, literature data on PACAP38 and PAC1R expression are only available for the Y-79 cell line. Olianas and coworkers noted that around 60% of Y-79 cells express membrane-bound PAC1R, and PACAP induces a concentration-dependent increase in adenylyl cyclase activity, with PACAP38 being six-fold more potent than PACAP27 [65]. In contrast, we observed the absence of membrane PAC1R in retinoblastoma, highlighting a significant distinction between the in vitro model and the clinical manifestation. Thus, the Y-79 cell line, which is mostly used for in vitro retinoblastoma studies, has the limitation of representing patients’ retinoblastoma. In the future, we need further studies to understand the precise reason for this.

In the second part of our experiment, we observed that nanomolar (100 nM and 500 nM) and 1 µM PACAP38 concentrations had no effect on the viability of Y-79 human retinoblastoma cells. In their investigation, Wojcieszak and Zawilska extensively explored how PACAP influences the viability of Y-79 cells derived from human retinoblastoma. Nanomolar (0.1–100 nM) concentrations of PACAP38 did not affect the viability of Y-79 cells, while micromolar (1–5 μM) concentrations of PACAP38 induced a dose-dependent decrease in tumor cell viability. The administration of a PAC1R antagonist, PACAP6-38, did not terminate this cytotoxic effect of PACAP38; furthermore, PACAP6-38 alone, in the same micromolar concentration, also produced a dose-dependent decrease in tumor cell viability. Micromolar concentrations of PACAP27 (0.1–5 μM) and a high-affinity selective PAC1 receptor agonist, maxadilan (1–2 μM), did not significantly affect the viability of Y-79 human retinoblastoma cells. [Disc^6^]PACAP38 and FITC-Ahx-PACAP11-38, two membrane-penetrating PACAP38 analogs which are inactive in PAC1, VPAC1, and VPAC2 receptors, also decreased the viability of Y-79 cells but with lower potency than PACAP38. This suggests that PACAP can exert its cytotoxic effects in a membrane receptor-independent way [66]. Our results are in accordance with the aforementioned study [66], showing that concentrations of PACAP38 up to 100 nM had no effect on the viability of Y-79 cells. PACAP38 proved to be cytotoxic only when used in micromolar (2 µM and 6 µM) concentrations. Despite this, it is important to emphasize that Y-79 is just one of numerous human retinoblastoma cell lines, necessitating future testing of PACAP38 effects across diverse in vitro models.

The activation of PACAP receptors in specific neoplasms can lead to growth stimulation, whereas in others, it results in inhibitory effects. These effects are influenced by various factors, such as the species’ origin, tumor type and origin, stage of differentiation, or the tumor environment [52,54]. For instance, in glioblastoma cases, the administration of PACAP27 resulted in the increased proliferation of mouse C6 glioma cells, while applying both PACAP isoforms led to a significant decrease in proliferation for T98G human glioma cells. In other human glioblastoma cell lines (M059K and M59J), PACAP agonists reduced cancer cell migration without affecting their proliferation. Furthermore, it has been demonstrated that the impact of PACAP is also dependent on the conditions within the tumor microenvironment [67]. On the other hand, there were instances where both pro-survival and anti-survival effects of PACAP38 were absent, as observed in JAR cytotrophoblast cells exposed to methotrexate treatment [68], and likewise in hepatocellular carcinoma cells (HEP-G2) [69]. The multifaceted nature of PACAP’s impact is highlighted, elucidating that opposite effects can emerge within the same cell line depending on factors such as exposure time, like in the case of LNCaP human prostatic tumor cells [70], or concentration, like in the case of Y-79 human retinoblastoma cells [66]. In human Y-79 retinoblastoma cells, the cytotoxic effects of PACAP38 were observed at concentrations equal to or exceeding 2 μM. However, the exact mechanism of PACAP38-induced Y-79 cell cytotoxicity is still unknown. It was observed that PACAP38 exerts its cytotoxic effects in a PAC1/VPAC1-2 membrane receptor-independent way in Y-79 cells without the activation of PKA, PKC, MEK1/2, p38, and JNK kinases [66]. Previous investigations have also indicated that PACAP38 acts as an intracrine factor, exhibiting the capacity to penetrate the internal cell compartment through direct translocation and endocytosis. This phenomenon results in a significant upsurge in the intracellular fraction, particularly at micromolar concentrations. Moreover, intracellular PACAP38 is not entirely degraded by intracellular enzymes and is able to activate intranuclear PAC1Rs [71,72]. The perinuclear PACAP38 and PAC1R positivity in human retinoblastoma detected by our first experiment could confirm these in vivo findings.

PACAP has demonstrated favorable outcomes in numerous pathological conditions which are primarily attributed to its cell-protective, antioxidant, and anti-inflammatory properties [34,35]. On the other hand, as indicated above, the PACAPergic system is affected by numerous malignant transformations as PACAP might exhibit both stimulatory and inhibitory impacts on tumor growth or cancer cell migration depending on the type of tumor [52,54]. This implies the possible exploration of developing selective agonists or antagonists for PAC1R, as well as analogs or antagonists for PACAP38, as valuable tools for diverse approaches to cancer treatment [52,73,74]. While in most diseases, PACAP or PAC1R agonist/antagonist treatment may pose significant limitations due to inadequate in vivo stability or limited penetration through the blood–brain barrier [75], in ophthalmic diseases, local application can overcome these factors. Previous in vivo studies have confirmed that PACAP38, when administered as eye drops, can permeate ocular barriers and exhibit retinoprotective effects [76]. Additionally, PACAP38 has been identified as a new candidate medication for Dry Eye Disease [77]. A study suggests that VIP and PACAP analogs, explicitly developed for therapeutic purposes, can modulate molecular and cellular processes relevant to treating high-risk neuroblastoma [78]. In our study, we proved that human retinoblastoma expresses PAC1R, and it is also known that in Y-79 human retinoblastoma cells, the micromolar concentration of PACAP38 exerts cytotoxic effects. These data highlight the therapeutic potential of PACAP38 and its analogs in retinoblastoma therapy. The results of Boisvilliers et al. [78] especially strengthen this potential in the case of retinoblastoma (even in its rare form, where retinoblastoma develops in the absence of an *RB1* mutation as a consequence of the somatic amplification of the *MYCN* gene), as they found that these peptide analogs are capable of inducing a sustained decrease in n-MYC expression and hold potential not only for neuroblastoma therapy but also for addressing other tumors. In our study, human retinoblastoma was found to express PACAP38 and PAC1R exclusively in a perinuclear dot-like pattern, with PACAP38 demonstrating cytotoxicity at concentrations of 2 µM and above in Y-79 retinoblastoma cells. Nevertheless, our studies also have limitations. To begin with, the limited number of cases prevents us from making a broad generalization that all retinoblastomas exhibit a uniform expression pattern. None of the cases in our study showcased fleurettes or neuronal differentiation which constituted more than half of the tumor, thereby hindering our ability to establish the immunoprofile for grade 1 tumors. Additionally, we cannot currently conclude whether PACAP signaling is implicated in the carcinogenesis of retinoblastoma. These aspects should be investigated more thoroughly in future studies. Furthermore, it is crucial to emphasize the need for exploring PACAP38 and its analogs in diverse in vitro models, establishing the groundwork for potential in vivo studies.

## 5. Conclusions

Elucidating the cellular origin of human cancers, including retinoblastoma, and understanding how cellular/subcellular context influences the probability of cancer initiation and progression would help to develop novel preventive and early-intervention therapies or improve existing ones. PACAP is involved in dozens of physiological and pathophysiological processes, and an increasing amount of evidence suggests that PACAP has diagnostic and therapeutic potential in certain diseases. In our study, we described, for the first time, the distribution of PACAP38 and PAC1R immunoreactivity in human retinoblastoma and we confirmed the cytotoxic effect of micromolar PACAP38 concentrations in human retinoblastoma cells. The facts that (i) PACAP38 and PAC1R are present in human retinoblastoma and that (ii) PACAP38 and its analogs had a cytotoxic effect on retinoblastoma cells suggest the potential role of PACAP38 and its analogs in retinoblastoma therapy.

## Figures and Tables

**Figure 1 life-14-00185-f001:**
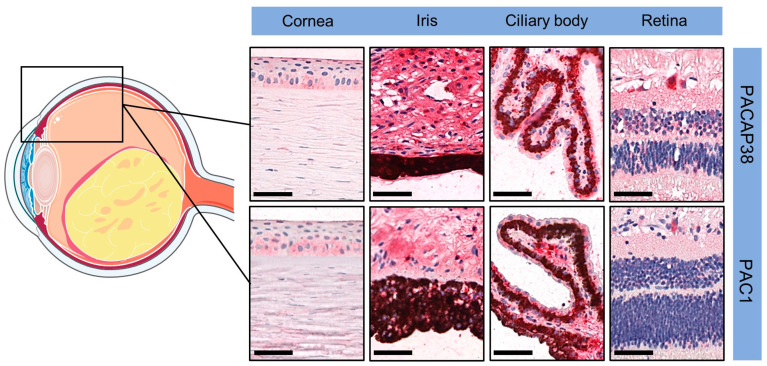
Representative pictures of PACAP38 and PAC1R immunopositivity in disease-free parts of the eyes (magnification 200×; scale bars: 50 µm). Certain parts of the figure were created using images from Servier Medical Art. Servier Medical Art by Servier is licensed under a Creative Commons Attribution 3.0 Unported License (https://creativecommons.org/licenses/by/3.0/, accessed on 1 November 2023).

**Figure 2 life-14-00185-f002:**
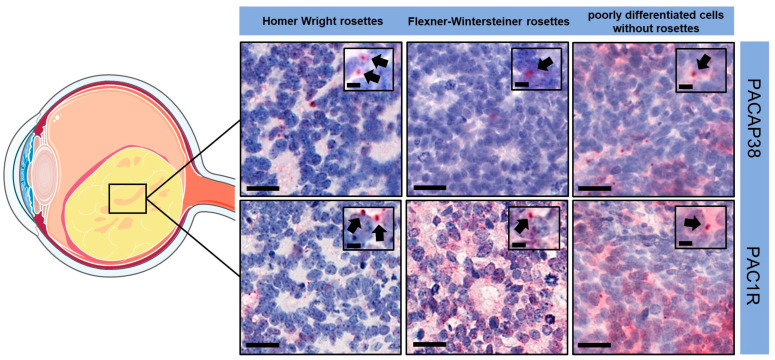
PACAP38 and PAC1R immunopositivity in the tumor samples (magnification: large pictures 400×, scale bars: 20 µm; index pictures 700×, scale bars: 5 µm). Black arrows indicated perinuclear, dot-like immunopositivity in retinoblastoma cells. The illustration includes components that were drawn utilizing images sourced from Servier Medical Art. Servier Medical Art by Servier is licensed under a Creative Commons Attribution 3.0 Unported License (https://creativecommons.org/licenses/by/3.0/, accessed on 1 November 2023).

**Figure 3 life-14-00185-f003:**
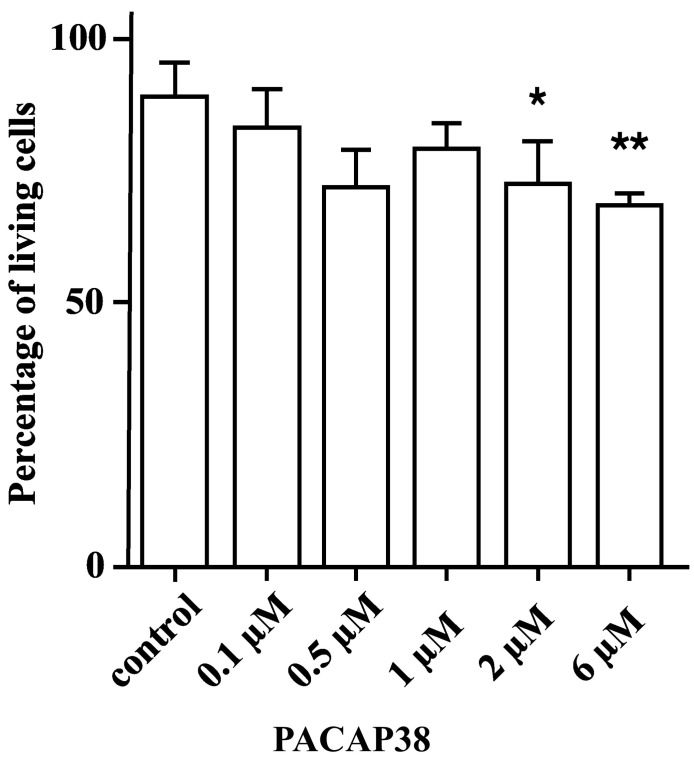
The percentage of living Y-79 cells after different concentrations of PACAP38 administration. (* *p* < 0.05; ** *p* < 0.01 compared to the control group. The data are presented as mean ± SD).

**Table 1 life-14-00185-t001:** Main clinical and demographic features of retinoblastoma cases.

Case	Sex	Age at Enucleation (Months)	Eye Involved	Tumor Site	Number of Tumor Foci	*RB1* Mutation
1	male	9.9	right	superotemporal	monofocal	no
2	female	16.6	right	superotemporal	multifocal	no
3	male	9.3	right	superotemporal	monofocal	no
4	male	4.1	right	adjacent to the optic disc	monofocal	yes
5	male	36.3	left	whole eye	monofocal	no
6	male	19.9	left	whole eye	monofocal	no
7	male	18.3	right	whole eye	monofocal	no

**Table 2 life-14-00185-t002:** Pathological findings of retinoblastoma cases (Mib-1: cell proliferation marker—percentage of immunoreactive tumor cells; n.d.: no data).

Case	Histomorphology	Necrosis	Calcification	Mib-1 (%)	Stage
1	tumor cells arranged in sheets, nests, and trabeculae (G4)	large	focal	80	pT2
2	tumor with many Homer Wright rosettes (G2)	small	focal	90	pT2
3	tumor with many Homer Wright rosettes (G2)	small	focal	90	pT2
4	tumor with many Flexner–Wintersteiner rosettes (G2)	large	no	50–60	pT2
5	tumor with occasional Homer Wight rosettes (G3)	large	focal	80	pT2
6	tumor cells arranged in sheets, nests, and trabeculae (G4)	large	focal	n.d.	pT2
7	tumor cells arranged in sheets, nests, and trabeculae (G4)	large	no	n.d.	pT2

## Data Availability

The data presented in this study are available on request from the corresponding author. The data are not publicly available due to ethical restrictions.

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
