# Peer review of "Investigation of PACAP38 and PAC1 Receptor Expression in Human Retinoblastoma and the Effect of PACAP38 Administration on Human Y-79 Retinoblastoma Cells"

_life, 2024, doi:10.3390/life14020185_

Round 1

Reviewer 1 Report

Comments and Suggestions for Authors

Dear authors, 

you presented a really interesting article regarding molecular pathways implied in human retinoblastoma. I think that the article is well written, methods are well specified and that it could have significant clinical resonance in the next years. May I suggest to shorten the introduction a bit? Although exhaustive, it's quite long when compared to the rest of the article.

Good job,

kind regards

Comments on the Quality of English Language

Only minor checks for english language are needed.

Author Response

We would like to express our gratitude to the Reviewer for dedicating her/his time to thoroughly read the article and for providing valuable and constructive feedback. We agree that the introduction section needs to be shortened, as you and Reviewer 3 suggested. Therefore, we removed parts from the introduction and incorporated them into the extended discussion section. We also conducted a spell check of the manuscript.

The changes implemented in the manuscript are apparent through the Microsoft Word track changes function (and we marked the text in red), while the rephrased sections, as recommended by the Assistant Editor, are indicated with a yellow background.

Reviewer 2 Report

Comments and Suggestions for Authors

In this manuscript, authors demonstrated that they found human retinoblastoma cells showed only perinuclear, dot-like immunopositivity for both PACAP38 and PAC1R irrespective of laterality, genetic background or histopathological features. The manuscript is attractive. 

1.  authors illustrated micromolar (2 µM and 6 µM) PACAP significantly decreased the tumor cell viability. how about other concentration, such as 1 µM or 10 µM. 

2. authors said that their work highlighted the potential role of PACAP38 and its analogues in retinoblastoma therapy. However, the introduction and discussion are not deeply described. Authors need more references to discuss. 

3. The English writing, unit format and references format should be carefully checked before submit.

Comments on the Quality of English Language

minor revison 

Author Response

We want to express our sincere gratitude for taking the time to review our manuscript and providing valuable comments and suggestions. Your input has been very helpful in improving the quality of our work. The changes implemented in the manuscript are apparent through the Microsoft Word track changes function (and we marked the text in red), while the rephrased sections, as recommended by the Assistant Editor, are indicated with a yellow background.

  1. Authors illustrated micromolar (2 µM and 6 µM) PACAP significantly decreased the tumor cell viability. How about other concentration, such as 1 µM or 10 µM. 

In our experiment, we started with the assumption that, according to previous literature, there was a difference between nanomolar and micromolar concentrations. We utilized 1 µM as a standard and assessed its effects at one-tenth (0.1 µM) and half (0.5 µM) concentrations. Subsequently, we doubled the standard dose (2 uM), and then tripled this double dose (6 uM). In the experiment conducted by Wojcieszak and Zawilska at the nanomolar level, they only investigated the range of 0.1nM -0,1 µM, while we examined 0.1 and 0.5 µM. Additionally, the former pair of authors explored concentrations between 1-5 µM, whereas we also investigated 6 µM. Based on our results, it can be stated that, supplementing the findings of the previous authors, the cytotoxic effect of PACAP38 on Y-79 retinoblastoma cells is observed from 2 µM. We did not investigate 10 µM, as we believed that the examination of previous data and the concentrations, we applied would be sufficient to determine the threshold of PACAP38 cytotoxicity on this cell line.

  1. Authors said that their work highlighted the potential role of PACAP38 and its analogues in retinoblastoma therapy. However, the introduction and discussion are not deeply described. Authors need more references to discuss. 

We agree with the Reviewer's point that we did not extensively discuss the clinical potential of PACAP and its analogs. To address this, we have dedicated the last paragraph of the Discussion section to this issue, incorporating new references.

  1. The English writing, unit format and references format should be carefully checked before submit.

We have thoroughly reviewed the manuscript to ensure the correctness of English writing, unit format, and references.

Reviewer 3 Report

Comments and Suggestions for Authors

Dear Authors,

Introduction:
- It should be shorten in 3 paragraph (currently it is too long): What is retinoblastoma - What are the genes / receptors changes - What is the aim of Your research.

Methods:
- Could the limited number of cases affected the findings of the study?

Discussion:
- It should focus more on retinoblastoma with brief comparison with PACAP38 and PAC1R in other tumors.
- The clinical impact of the findings should be better explained and discussed in view of the following.
The expression of PACAP38 and PAC1R has been reported  in disease-free areas of the eye (lines 207-212), and then "we were not able to compare the PACAP38 and PAC1 receptor immunoprofile of "normal" and tumor tissue" (lines 264-265). Also there is "no or only weak PACAP38 and PAC1 receptor signaling in inner and outer nuclear layer " (line 268-273). 

Author Response

Introduction: - It should be shorten in 3 paragraph (currently it is too long): What is retinoblastoma - What are the genes / receptors changes - What is the aim of Your research.

We would like to express our gratitude to the Reviewer for dedicating her/his time to thoroughly read the article and for providing valuable and constructive feedback. The changes implemented in the manuscript are apparent through the Microsoft Word track changes function (and we marked the text in red), while the rephrased sections, as recommended by the Assistant Editor, are indicated with a yellow background.

We agree that the introduction section needs to be shortened, as you and Reviewer 1 suggested. Therefore, we removed parts from the introduction and incorporated them into the extended discussion section.

Methods: - Could the limited number of cases affected the findings of the study?

Certainly, one limiting factor of our study is the small sample size. We acknowledge that the case number is limited; however, given the low incidence of retinoblastoma in our region, these were all the cases we were able to identify. We placed the limitations of our study at the end of the conclusion chapter with specific attention to the Reviewer's valuable feedback.

Discussion:

- It should focus more on retinoblastoma with brief comparison with PACAP38 and PAC1R in other tumors.

As requested by the Reviewer, we have expanded the Discussion section.

- The clinical impact of the findings should be better explained and discussed in view of the following.

We agree with the Reviewer's point that we did not extensively discuss the clinical potential of our findings. To address this, we have dedicated the last paragraph of the Discussion section to this issue, incorporating new references.

The expression of PACAP38 and PAC1R has been reported in disease-free areas of the eye (lines 207-212), and then „we were not able to compare the PACAP38 and PAC1 receptor immunoprofile of „normal” and tumor tissue” (lines 264-265). Also there is „no or only weak PACAP38 and PAC1 receptor signaling in inner and outer nuclear layer „ (line 268-273). 

Thank you for pointing out, and we agree with the Reviewer that the specified section was challenging to interpret, and we expressed our concepts inaccurately. Addressing this, we completely rewrote the relevant portion for clarity.

Round 2

Reviewer 3 Report

Comments and Suggestions for Authors

Dear Authors,

Thanks for this revision.

Author Response

We thank the reviewer for reviewing our revised manuscript and for the valuable and insightful advice provided during the review process.